# Validation of a Questionnaire to Assess Smoking Habits, Attitudes, Knowledge, and Needs among University Students: A Pilot Study among Obstetrics Students

**DOI:** 10.3390/ijerph182211873

**Published:** 2021-11-12

**Authors:** Laura Campo, Francesca Vecera, Silvia Fustinoni

**Affiliations:** 1Environmental and Industrial Toxicology Unit, Fondazione IRCCS Ca’ Granda Ospedale Maggiore Policlinico, 20122 Milan, Italy; silvia.fustinoni@unimi.it; 2EPIGET—Epidemiology, Epigenetics, and Toxicology Lab, Department of Clinical Sciences and Community Health, Università degli Studi di Milano, 20122 Milan, Italy; francesca.vecera@studenti.unimi.it

**Keywords:** tobacco smoke, e-cigarette, HTP, questionnaire, validation, undergraduates, students, passive smoke

## Abstract

In Italy, smoking is still widespread among a relatively high percentage of young people. This study aimed to develop and validate a questionnaire to assess smoking habits, passive smoke exposure, electronic cigarette (e-cig) and heated tobacco product (HTP) use, attitudes, knowledge, and needs among undergraduates. A questionnaire consisting of 84 items was developed starting from a literature review and existing questionnaires. A two-round validation was performed by a team of 10 experts. The item-level content validity index (I-CVI), the scale-level content validity index (S-CVI), and the kappa statistics k, taking into account chance agreement, were calculated from the experts’ rating. The questionnaire was emailed to 114 students from the Obstetrics Degree of the University of Milan (Italy) to be pilot tested. After the second round of validation, all indexes were above the respective acceptability criteria: the I-CVI was 1.00 for all but three items, k was >0.74 (“excellent”) for all items, and the S-CVI was 0.964. Eighty-nine students participated in the survey: 17 classified themselves as smokers, eight as new product users, and four as former smokers, 72% students declared to routinely spend free time with smokers, while almost all students believed that healthcare professionals play a pivotal role in preventing smoking towards their patients and society. This questionnaire will be used in a survey among students from the University of Milan as a first step for future campaigns targeting health promotion.

## 1. Introduction

Tobacco smoking represents a dramatic public health problem worldwide, being the direct cause of about 6 million deaths a year [1]. Active smoking, as well as passive smoking, is classified by the International Agency for Research on Cancer (IARC) as a Group 1 human carcinogen. Cigarette smoking is considered one of the most important preventable risk factors for the development of chronic diseases and is associated with several adverse health effects, such as asthma, respiratory tract infections, COPD, heart attack, and stroke [2].

In the frame of the WHO Tobacco Control Convention [3,4], several initiatives have been implemented in Italy, including the activation of clinics for the cessation of tobacco smoking, the establishment of the toll-free number against smoking, and the issuing of new regulations imposing a ban on smoking in public places [5], which were then extended to the external premises of schools and campuses [6] and of the pediatric, obstetric, and gynecological departments and within vehicles in the presence of children and pregnant women [7].

The adoption of these measures, and in particular the regulatory measures, have had positive effects. Indeed, the percentage of smokers in Italy is constantly decreasing, going from 30% in 2001 to 22% in 2019 [8]. The same is not observed among young Italians, among which the prevalence of smokers is over 30%, one of the highest in Europe [9]. Tobacco smoking is also relevant among undergraduates, with percentages of smokers in the 12–49% range, even among medicine and health profession undergraduates [10,11,12].

These data therefore highlight the need for health education interventions for the training of students and future health professionals. In particular, it is of primary importance to implement policies that involve educational institutions, by focusing on prevention, on promoting smoking cessation, and protecting non-smokers from passive smoking.

The University of Milan (Italy), also known as “La Statale”, is one of the biggest universities in Italy, with about 64,000 students enrolled in 10 different faculties. In November 2019, a new regulation has been implemented at “La Statale”, banning smoking of both traditional tobacco cigarette and electronic cigarettes or heated-tobacco-products (HTP) devices in all external premises of the campus. However, some studies indicate that policies that incorporate prevention and cessation programming produce better results in terms of reducing smoking behavior among undergraduates [13].

To investigate the possible areas of intervention in anti-smoking policies, the University of Milan implemented the project “LA STATALE SMOKEFREE” in 2020. The first step of the project is the submission to all students of a questionnaire to investigate their smoking habits, exposure to passive smoke, compliance with smoking bans on campus, and the attitudes of smokers and non-smokers towards the anti-smoking policies. This survey will allow to obtain an updated picture of smoking behaviors and attitudes useful for addressing new prevention campaigns or improved actions of the anti-smoking policy at “La Statale”.

The aim of the study here presented was the development and validation of the questionnaire to be used for the “LA STATALE SMOKEFREE” project. As a step in the validation process, a pilot study among students from the bachelor study program for obstetrics was conducted.

## 2. Materials and Methods

The questionnaire was developed according to the recommendation presented by the Association for Medical Education in Europe (AMEE) in the “Developing questionnaires for educational research: AMEE Guide No. 87” [14]. The Guide presents a systematic, seven-step process for designing high-quality questionnaires, including conduct a literature review, conduct interviews and/or focus groups, synthesize the literature review and interviews/focus groups, develop items, conduct expert validation, conduct cognitive interviews, and conduct pilot testing.

### 2.1. Literature Search

A literature search was performed in the Medline database through Pubmed and using various combination of the key-words “smoking”, “questionnaire”, “tobacco smoke”, electronic cigarettes”, “students”, “young people”, and “survey”. The search was limited to studies involving Italian students. Moreover, the websites of some institutional and international bodies were consulted, including the World Health Organization (WHO), the US Center for Disease Control and Prevention (CDC), and the Italian National Institute of Health (ISS).

### 2.2. Interviews

Interviews were conducted among students (non-smokers, smokers, and e-cig or HTP users) conducting their thesis work at the Environmental and Industrial Toxicology Unit to obtain inputs from prospective respondents. The interviews covered all topics to be included in the questionnaire. Students from the degree courses in chemical safety and toxicological environmental sciences, biomedical laboratory technics, and health professions of prevention sciences, for a sum of 9 students, were involved in this step.

### 2.3. Summary of Literature Search and Interviews

The full texts of the retrieved papers were read, and the information was synthetized in tabular form highlighting the characteristics of the studies, i.e., authors, year and place of studies, number and type of students involved, type of questionnaire used for the research, and results.

The information deriving from interviews was summarized by making list of the terminology used by students while talking freely about the proposed topic. More focused questions were asked about passive smoke exposure, e-cig and HTP use, attitudes, and educational needs.

### 2.4. Item Development

As a starting point for the item development, we used a questionnaire previously developed and used in our laboratory to assess smoking habits in the general population [15]. The existing questionnaire was then adapted to the population to be studied based on the literature search and the interviews conducted among students.

### 2.5. Questionnaire Validation

The content validity was assessed by conducting a two-round validation [16,17,18]. The experts participating in the validation process were professionals in the field of hygiene and public health, medical statistics, social statistics, pedagogy, and smoking habit prevention from the University of Milan and from Fondazione IRCCS Ca’ Granda Ospedale Maggiore Policlinico (Milan, Italy). The first round of the validation process was conducted by 10 experts, while for the second round a subgroup of five experts was chosen, eliminating experts who were consistently lenient or consistently harsh, and including those experts that had given useful comments about the items in round one [18]. Each expert was asked to rate each item in terms of its relevance to the underlying construct using a 4-point rating as follows: 1 = not relevant, 2 = somewhat relevant, 3 = quite relevant, 4 = highly relevant.

The item-level content validity index (I-CVI) and the scale-level content validity index (S-CVI) were used to assess the content validity of each item and of the questionnaire as a whole, respectively [17,18]. Based on the experts’ ratings, I-CVI was calculated as the proportion of experts rating 3 or 4 the item relevance. The scale-level content validity index (S-CVI) was calculated both with the universal agreement calculation method (S-CVI/UA) (proportion of items that achieves a relevance rating of 3 or 4 by all the experts) and with the averaging calculation method (S-CVI/Ave) (average of the I-CVIs for all items on the scale). The acceptability criteria for these indexes are I-CVI > 0.78, S-CVI/UA > 0.80, and S-CVI/Ave > 0.90. Moreover, to take into account the chance agreement, a modified Cohen kappa statistic, k*, was computed as follows:$$k^{*}=\frac{\left(\mathrm{I-CVI}-\mathrm{Pc}\right)}{\left(1-\mathrm{Pc}\right)}$$
where Pc is the probability of chance agreement, calculated as follows:$$\mathrm{Pc}=\left[\frac{N!}{A!\left(N-A\right)!}\right]*\;0.5^{N}$$
where N is the number of experts and A is the number of expert rating 3 or 4.

Evaluation criteria for k* are: fair = 0.40 < k* < 0.59; good = 0.60 < k* < 0.74; excellent = k* > 0.74 [17,18].

In addition to content validity, experts were also asked to rate the clarity of each items using a 4-point scale (1 = not clear, 2 = somewhat clear, 3 = quite clear, 4 = clear) and to supply suggestions or comments to modify the items, if necessary.

### 2.6. Cognitive Interviews

Students involved in the former interview step were involved also in cognitive interviews. The validated questionnaire was submitted to students in a paper-based format and their response to items was discussed to be sure that the used terminology was clear and that the included anchors were exhaustive. The verbal probing technique was used, whereby the interviewer asked specific questions about items while the respondent was answering each questions (i.e., “How did you arrive at that answer?”, “Is this terminology clear to you?”, “Would you add other anchors?”, and so on).

### 2.7. Pilot Study

The students from the Obstetrics Degree of the University of Milan, Italy, participated in the pilot study. The course is part of the Faculty of Medicine and has a duration of three years. In the academic year 2020–2021, there were 45, 37, and 32 students enrolled in the course in the first, second, and third years, respectively, for a sum of 114, including 111 females and 3 males.

Using the institutional e-mail address of students, an invitation letter, containing the link to access the consent form and questionnaire, was emailed to students on 21 December 2020. In the following weeks, the students received two reminds, respectively on 11 and 18 January 2021. The access to the questionnaire was possible for students until 24 January 2021.

The students interested to participate in the study expressed their informed consent by clicking on the appropriate box. Only after giving their consent, students had access to the questionnaire and were therefore able to complete it. A copy of the informed consent signed was accessible by connecting to a dedicated link. Students could complete the questionnaire even only partially, and then resume completing it or modify the responses already given.

The study was approved by the ethics committee of the University of Milan (project code 82/20).

### 2.8. Data Treatment and Statistical Analysis

Before statistical processing, collected data were anonymized by the University Teaching and Learning Innovation and Multimedia Technology Centre (CTU), i.e., the link between the answers provided by the students and their identity was eliminated. Data were provided by CTU in the form of a Microsoft Excel file without any link between the identity of the responders and the corresponding questionnaire responses. Descriptive statistical analysis was performed using Microsoft Excel (Microsoft Office Professional Plus 2016 for Windows 10 Pro).

For each item, frequency was calculated for each response’s anchor, while mean, minimum, and maximum were calculated where appropriate.

## 3. Results

### 3.1. Summary of Literature Search and Interviews

We found 31 papers relative to the topic of smoking and Italian colleges, spanning from 1988 and 2020 (results not shown). Only two studies investigated degree courses different from Medicine and Health professions. Among the retrieved study, 19 stated to use a validated questionnaire, and in particular, eight studies used the WHO Global Health Professions Student Survey (GHPSS) [19]. Only one study reported the validation of a questionnaire related to the use of e-cigs [20], while no study reported items on attitudes toward legislation and educational needs.

As regards the research conducted in web sites of institutional and international bodies, two questionnaires were considered in particular: the GHPSS and the ISS survey. The first was developed by WHO in collaboration with the US Center for Disease Control and the Canadian Public Health Association, to collect data on tobacco use and knowledge of smoking cessation techniques among health professional students from all WHO member states participating in the tobacco control program [19], while the second is a survey conducted yearly from the Italian National Institute of Health (ISS) and taking into account various aspects of tobacco smoking with updates on the latest trends [8].

From interviews, it emerged that the terminology used by students was in line with the literature. In addition, we received suggestions useful to formulate the items regarding in particular the occasions of passive exposure, the behaviors of smokers and nonsmokers on campus, the appeal and negative aspects of e-cigs and HTPs, and educational needs and possible initiatives to be undertaken by the University of Milan.

### 3.2. Item Development

Based on our previous experience [15] and results from the literature search and the interviews, a questionnaire with seven sections and 84 items was developed: demographics (Appendix A), current smoking of traditional tobacco cigarettes (Appendix A), past use of traditional tobacco cigarettes (Appendix A), current or past use of electronic cigarettes or HTP products (Appendix A), passive smoke exposure (Appendix A), awareness of health issue related to smoking (Appendix A), knowledge and attitudes towards smoking legislation, and educational needs (Appendix A).

Appendix A included four items on sex, age, year of course attendance, and a smoking declaration defined as follows: never smoker (i.e., never smoked or have smoked less than 100 cigarettes in life), past smoker (current nonsmoker, but a person who has smoked more than 100 cigarettes in life), current smoker, current user of electronic cigarette or HTP products exclusively, current dual users (both traditional and electronic cigarettes or HTP products). Based on this smoking declaration, participants were linked to different sections of the questionnaire (i.e., never smoker completed Appendix A, past smokers Appendix A, current smokers Appendix A, current user of electronic cigarette or HTP products Appendix A, and dual users Appendix A).

Appendix A contained 12 items related to smoking of traditional tobacco cigarettes, such as age of initiation, reason for initiation, use at university, intention to quit, and six items from the Fagerström test to evaluate nicotine dependence (these six questions were not validated by experts) [21]. Appendix A included seven items related to the past use of traditional tobacco cigarettes, such as age of initiation and of cessation, reason for initiation and for cessation, and methods used to quit smoking. Appendix A investigated the current or past use of electronic cigarettes (11 items) or HTP products (10 items), such as age of initiation, reason for initiation, use at University, intention to quit. Appendix A included nine items related to ETS exposure, such as opinions on health effects of ETS exposure to traditional cigarettes or electronic cigarettes or HTP products, living with smokers, perception and circumstances of ETS exposure, and smoking ban at home. Appendix A contained 12 items related to awareness of health effects of active and passive smoking, opinions on electronic cigarettes and HTP products, and eight items related to the role of health professionals in helping patients to quit smoking derived from the GHPSS. Finally, Appendix A included 11 items on knowledge and attitudes towards Italian legislation on smoking, and educational needs at University (Appendix A).

The complete questionnaire is reported in Appendix A.

### 3.3. Questionnaire Validation

Table 1 reports the results of the two-round validation. In the first-round validation, I-CVIs was in the 0.7–1.0 range, with 50/84 items with I-CVI 1.0, 24 with I-CVI 0.9, seven with I-CVI 0.8, and three with I-CVI 0.7, the latter slightly lower than the acceptability criteria (0.78). Considering the k* index, the I-CVIs evaluation was excellent (k* > 0.74) for 81 items and good (0.60 < k* < 0.74) for three items. S-CVI/UA and S-CVI/Ave were 0.595 (lower than the acceptability criteria of 0.80) and 0.943 (higher than the acceptability criteria of 0.90), respectively.

In light of these results, the items with I-CVI lower than 0.78 were not excluded from the following version of the questionnaire, as their k* value was found to be equal to 0.66, therefore “good”, but they were revised. We then proceeded to examine and possibly modify all the items based on the suggestions provided by the experts for those aspects related to clarity, such as the use of a homogenous terminology throughout the whole questionnaire, the use of closed-ended instead of open-ended items for time-related items, and adding the option “I don’t know” or “I don’t remember” as anchors to several items.

In the second round of the validation, I-CVIs was in the 0.8–1.0 range, with 81/84 items with I-CVI 1.0 and 3 with I-CVI 0.80. Considering the k* index, the I-CVIs evaluation was excellent (k* > 0.74) for all items. S-CVI/UA and S-CVI/Ave were 0.964 and 1.000, so higher than their respective acceptability criteria.

### 3.4. Cognitive Interviews

The students participating to the cognitive interviews were familiar with the terminology used in the validated questionnaires.

Some items were slightly modified following the cognitive interviews. For example, the response anchor “using smoke free-apps” was added to item C6 (“How did you manage to quit smoking traditional cigarettes?”) and the response anchor “openly” substituted the former anchor “freely” in item C6/D7/D17 (“When you are with your family, you smoke traditional cigarettes/you use e-cigs/you use HTPs”).

### 3.5. Pilot Study

We received valid responses by 89 students out of 114 receiving the invitation. Table 2 reports the main characteristics of the study participants together with the main results. Detailed results are reported in Appendix A and briefly described in Section 3.6.

Some items were slightly reviewed after the pilot study. We added response anchors to some closed-ended items after carefully examining free-text responses, specifically inserted in the questionnaire. These include item C7 (“Why did you quit smoking traditional cigarettes?”), “I didn’t like it”; item F7 (“What is the appeal of e-cig and HTPs in your opinion?”), “they are trendy”, and item F9 (“Which initiatives can help quit smoking or prevent young people from starting smoking in your opinion?”), “more information in school on the harm of smoking”.

Some time-related items (item C4 “How long have you been without smoking traditional cigarettes” and item D2/13 “How long have you been using e-cigs/HTPs”) were formerly open-ended items (“write years, months, days”), but in some cases the responses highlighted that these items had been misunderstood, so they were transformed in closed-ended items with a five-point scale. For the same reason, item D4 (“How many ml of liquids do you use per day?), formerly an open-ended item, was transformed in a closed-ended item with a four-point scale.

Two items regarding the awareness of the health effects of passive smoking were moved from Appendix A (“Passive smoking”) to Appendix A (“Awareness of the health effects of active and passive smoking”) (now items F5 and F6, formerly E1 and E2, respectively).

### 3.6. Pilot Study Population

#### 3.6.1. Study Participants

Appendix A reports the answers collected in Appendix A of the questionnaire. Ninety students, out of 114, accessed the questionnaire. The first call (21 December) was answered by 51% of the participants, the second (11 January) by 38% and the third (18 January) by 11%. Only one student, despite having accessed the study link, did not express their consent to participate in the study. Therefore, there were 89 (participation rate 78%) valid participants. The participation rate was similar among students enrolled in the different years of the course: 75%, 73%, and 81% for students in the first, second, and third years, respectively. Of the 89 participants, three were males and 86 females. Mean age was 22 years.

As regards the smoking declaration, 72% students classified themselves as never smoker, 4.5% as former smokers, 18% as current smokers of traditional cigarettes exclusively, 4.5% as current users of electronic cigarettes or HTP only, and 1% as dual smoker.

#### 3.6.2. Active Smoking of Traditional Tobacco Cigarettes

Appendix A reports the answers collected in Appendix A. Mean age of initiation was 17 years, with peer pressure (76%) and pleasure (59%) provided as the main reasons for initiation. Most (53%) were non-daily smokers. Daily smokers smoked mostly less than 10 cigarettes/day. Most smokers (88%) smoked only outdoors, while 47% admitted to smoke not in the smoking area on campus, 59% has been advised by a doctor to quit smoking, but only 29% is planning to quit smoking in the next six months.

According to the Fagerström test, 88% of smokers were found to have a mild (total score 0–2) nicotine addiction.

#### 3.6.3. Former Smoker of Traditional Tobacco Cigarettes

Appendix A reports the answers collected in Appendix A. Mean age of initiation was 16 years, with stress being the main reason for initiation. All former smokers managed quit smoking without help and the main reason for quitting was concern for their own health.

#### 3.6.4. Electronic Cigarettes or HTP Users

Appendix A reports the answers collected in Appendix A. Only three students declared to be current users of electronic cigarettes, while 10 students declared to be past users and most students (85%) to have never tried it. Considering current and past users, most of them started using e-cig as an alternative to traditional cigarettes or as an aid to quit traditional cigarettes (38%), but most of them (46%) admit they have started or gone back to smoking traditional cigarettes after starting using electronic cigarettes.

As regards HTPs, five students declared to be current users and six past users, while most students (78, 88%) have never tried it. Considering current and past users, most of them have started using HTP as an alternative to traditional cigarettes (55%) and most of them have given up traditional cigarettes after starting using HTP.

#### 3.6.5. Passive Smoking

Appendix A reports the answers collected in Appendix A. The great majority of students (94%) consider passive smoking of traditional tobacco cigarettes bad for health, while a lower percentage, 76%, consider passive smoking of e-cig or HTP bad for health. Exposure to passive smoking is a common experience for students: 30% had been exposed in the previous week, 44% lived with smokers, and 72% usually spent leisure time with smokers. Most (63%) were exposed only outdoors, but 19% also indoors, mainly at home (70%). A total smoking ban was present in many houses (47%).

#### 3.6.6. Awareness of Smoking Health Related Issues and Role of Healthcare Professionals

Appendix A reports the answers collected in Appendix A. All students were aware of health risks associated with active smoke of tobacco cigarettes, and the great majority think that active smoke of e-cigs or HTP is dangerous for health. E-cigs and HTP are found appealing mostly because they have different flavors (48%), but also because they may be used where traditional cigarettes cannot be smoked (43%). However, most students think that e-cigs and HTPs may introduce nonsmokers to smoke. A smoking ban in outdoors area (52%) and informative advertising on the harm of smoking (48%) could help users to quit smoking.

As regards the role of healthcare professionals, 71% of students believe that healthcare professionals serve as role models for their patients and society regarding smoking habits. The overwhelming majority of students believe that healthcare professionals should regularly advise their smoking patients to stop smoking traditional cigarettes or new products (≥85%) and they should receive specific training on smoking cessation techniques. Moreover, 48% of the participants believe that healthcare workers who smoke are less likely to advise patients to stop smoking. Regarding university education, 91% of students have learned that it is important to record a history of tobacco use as part of a patient’s medical history, but 72% have never received training on the approaches to quit smoking to be used with patients. Most participants are interested in receiving more information on the topic of “smoking” during their university studies.

#### 3.6.7. Knowledge and Attitudes towards Smoking Legislation, and Educational Needs

Appendix A reports the answers collected in Appendix A. Most students (≥79%) know the Italian legislation in force and its motivations, but only 56% are aware of the ban on smoking traditional cigarettes and e-cigs in the outdoor areas of schools and universities, only 40% are aware that the University of Milan has internal regulations on this matter, and 63% declared that the smoking ban in outdoor areas of the University is not complied with. Finally, 65% of students suggested that the University of Milan could offer smokers courses to help them quit smoking, 58% would like greater control over compliance with existing bans, and 48% information campaigns on the harm of smoking.

## 4. Discussion

In this study, we have developed, validated, and tested in a pilot study the questionnaire to be used in the project “LA STATALE SMOKEFREE”, aimed at all students from the University of Milan.

The core of the questionnaire was the questionnaire that we had used in our previous experience with an adult population [15], adapted to take into account the specificity of the population now considered. In particular, the developed questionnaire covers different areas of the “smoking” topic, also considering new smoking products (i.e., e-cigs and HTPs) quite popular among young people, the passive exposure to both traditional and new products, knowledge and attitudes towards national regulation, educational needs, and interest in initiatives undertaken by the University of Milan. As about 13% of students from the University of Milan are enrolled in the faculty of Medicine, the section dedicated to the knowledge of smoking health related issues was enriched with ten items addressed to healthcare professionals and derived from the WHO GHPSS [19]. Only students from the faculty of Medicine will have access to this set of very specific items.

Following the two-round validation, performed by a team of ten experts, no item was excluded from the final version of the questionnaire, as the parameters for the content validity were all higher than their respective criteria. We accepted several suggestions regarding the clarity of the items, mostly relevant to the uniformity of language throughout the questionnaire. Finally, we performed a pilot study to test the questionnaire in the field. The students from the Obstetrics degree of the University of Milan were chosen as a subgroup of the target population for three main reasons: they were students from the faculty of Medicine, so they also had to respond to items derived from the GHPSS survey; they were a relatively small sample (114 students enrolled); and they were supposed to be interested in the topic, thus ensuring a high participation rate.

The implementation of the pilot study was useful to review some items based on students’ responses and to receive inputs that had not emerged from the previous steps (interviews, expert validation, and cognitive interviews). Moreover, the pilot study allowed us also to test the planned delivery mode, which is a web-based format. In particular, we tested the mailing list, the functioning of several links connecting the responders to the different sections of the questionnaire, the receiving of completed questionnaires from CTU, and finally the data treatment (anonymization and database organization).

As expected, the participation rate was relatively high (78%). It is likely that these students are very susceptible to issues regarding tobacco smoking, as they receive information on the effect of cigarette smoking on the health of pregnant women and unborn children. As the same time, the two reminds to the first calls contributed to power the participation rate, passing from 51% at the first call to final 78% after the third call. As students receive many institutional communications via email and not all students regularly check their inbox, this strategy was helpful in keeping the survey alive and will be maintained for the project “LA STATALE SMOKEFREE” as well.

As far as we know, this is the first time that students from the Obstetric degree have been investigated about their smoking habits. The percentage of traditional tobacco cigarette smokers (19%) among participants was similar to the prevalence of cigarette smoking among Italians aged 15–24 years (20.7%) [8]. In comparison to other studies conducted in Italy among medicine or healthcare professionals, the percentage found here is much lower than that reported in studies carried out in other degree courses of the health professions (31–45%) [22,23], while it is in line with data obtained in the most recent studies carried out in the medicine degree (12–22%) [24,25]. About 5% of students declared to use exclusively electronic cigarettes or HTP products. This percentage is in line with national data (about 6%), but much lower than that reported in a recent study on students (20%) [20].

We found that most participants were exposed to passive smoke, especially from traditional cigarettes. Students, even if fully aware of the health risks posed by passive smoking, did not seem to be aware of their exposure. Indeed, only 30% acknowledged it. The underestimation of exposure may be linked both to a misunderstanding of exposure [26], but also to difficulties in self-assessment related to passive smoke duration and intensity, as we found in our previous studies in adults [19] and in children [27]. A complete smoking ban at home was reported by 48% of students. This percentage is lower than the 63% that we recently found in homes with children in Milan [27]. As students often live alone or with other students, it is likely that the rules at home are less strict than in homes with children.

The majority of students (71%) believe that healthcare professionals serve as role models for their patients and society regarding smoking habits. This percentage is higher than 57% obtained in a previous study conducted in 2009 in four European countries, including Italy [10], and may be indicative of a high involvement of these students in the topic. Students had a high knowledge of the legislation in force, but the same was not true for regulations concerning the school and university. This discrepancy underlines the need both for a better communication and for the introduction of prevention and cessation programming in anti-smoking policies [13]. In this regard, the study participants demonstrated to be interested in any initiatives that the University of Milan could undertake to help smokers quit smoking and to protect the health of non-smokers.

The strength of this study is that a comprehensive questionnaire has been developed and validated, covering several areas of such an important topic as smoking. The main limitation is that the student population chosen for the pilot study was particularly susceptible to the topic and it was made up almost exclusively of female students, so it could be not fully representative of the target population as regards smoking habits. However, the main interest of the pilot study was to test the questionnaire and bring out issues relevant to understanding items. Furthermore, it should be noted that the percentage of female students at the University of Milan is higher than that of male students (60% vs. 40%).

## 5. Conclusions

In conclusion, a questionnaire to investigate smoking habits, knowledge, and attitudes of students from the University of Milan was developed, validated, and tested in a pilot study. The implementation of the “LA STATAL SMOKEFREE” project will certainly provide further useful information to investigate the possible areas of intervention in order to obtain success in the prevention of smoking at the University.

## Figures and Tables

**Table 1 ijerph-18-11873-t001:** Results of the two-round validation.

	N Items Rated 3 or 4 by Experts	N Items with I-CVI > 0.78	S-CVI/UA	S-CVI/Ave	N Items with k* = Fair	N Items with k* = Good	N Items with k* = Excellent
1° round	50/84	81	0.595	0.943	0	3	81
2° round	81/84	84	0.964	1.000	0	0	84

Acceptability criteria: I-CVI > 0.78; S-CVI/UA > 0.80; S-CVI/Ave > 0.90. Evaluation criteria for k*: fair = 0.40 < k < 0.59; good = 0.60 < k < 0.74; excellent = k > 0.74 [17,18].

**Table 2 ijerph-18-11873-t002:** Main characteristics of the study participants and the main results of the pilot study.

Question, Statistics	Anchors	Response
sex, N (%)	Female	86 (97%)
	Male	3 (3%)
age (years), mean (min-max)		22 (19–38)
Do you currently smoke? N (%)	No, i.e., I have never smoked or have smoked less than 100 traditional cigarettes in my life	64 (72%)
	No, but I used to, i.e., I have smoked at least 100 traditional cigarettes in my life	4 (5%)
	Yes, I smoke traditional cigarettes only	16 (18%)
	Yes, I use electronic cigarettes (e-cigs) or heated tobacco products (HTPs) only	4 (4%)
	yes, I smoke both traditional cigarettes and e-cigs/HTPs	1 (1%)
Do you live with smokers? N (%)	Yes, and they smoke in my presence/at home	20 (22%)
	Yes, but they do not smoke in my presence/at home	19 (21%)
	No	50 (56%)
Have you been exposed to passive smoking over the last week? N (%)	Yes	27 (30%)
	no	62 (70%)
In your house, traditional cigarettes: N (%)	Are not allowed in any rooms	42 (47%)
	Are only allowed in some rooms	6 (7%)
	Are only allowed outdoors	37 (42%)
	Are allowed everywhere	4 (4%)
Is active cigarette smoking bad for your health? N (%)	Yes	89 (100%)
	Yes, but only in particular conditions	0
	No	0
	I don’t know	0
Is active e-cig smoking bad for your health? N (%)	Yes	76 (85%)
	Yes, but only in particular conditions	2 (2%)
	No	0
	I don’t know	11 (12%)
Is active HTP smoking bad for your health? N (%)	Yes	79 (89%)
	Yes, but only in particular conditions	0
	No	0
	I don’t know	10 (11%)
Is passive smoking from traditional cigarettes harmful to the health of non-smokers? N (%)	Yes	84 (94%)
	Yes, but only in particular conditions	4 (4%)
	No	0
	I don’t know	1 (1%)
Is passive smoking from e-cigs or HTPs harmful to the health of non-smokers? N (%)	Yes	68 (76%)
	Yes, but only in particular conditions	5 (6%)
	No	4 (4%)
	I don’t know	12 (13%)
Do healthcare professionals serve as role models for their patients and society in terms of smoking habits? N (%)	Yes	63 (71%)
	No	15 (17%)
	I don’t know	11 (12%)
What initiatives could the University of Milan undertake to help smokers quit smoking and protect the health of non-smokers? N (%)	Informative campaigns on the harm of smoking	43 (48%)
	Greater control over compliance with existing bans	52 (58%)
	Launching specific courses on smoking issues	27 (30%)
	Offering smokers courses to help them quit smoking	58 (65%)

## Data Availability

The data presented in this study are available in Appendix A.

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
