# Peer review of "Validation of a Questionnaire to Assess Smoking Habits, Attitudes, Knowledge, and Needs among University Students: A Pilot Study among Obstetrics Students"

_ijerph, 2021, doi:10.3390/ijerph182211873_

Round 1

Reviewer 1 Report

The present study has as its objective "".
In my opinion, information related to the development and validation of the questionnaire is mixed with data obtained from the population in which the pilot study has been carried out, for example, in part of the discussion I think it would be appropriate to eliminate the parts referring to the results of the pilot study.
In general, the text seems to me to be too long and poorly structured.
I would suggest that the authors structure the content of the sections of their article along the lines of the one presented in this paragraph:
"The questionnaire was developed according to the recommendation presented by the Association for Medical Education in Europe (AMEE) in the "Developing questionnaires for educational research: AMEE Guide No. 87" [14]. The Guide presents a systematic, seven-step process for designing high-quality questionnaires, including conduct a litera-ture review, conduct interviews and/or focus groups, synthesize the literature review and interviews/focus groups, develop items, conduct expert validation, conduct cognitive interviews, and conduct pilot testing."

Author Response

The present study has as its objective "". The aim of the study was the development and validation of the questionnaire to be used for the “LA STATALE SMOKEFREE” project. The developed questionnaire was pilot tested among students from the bachelor study program for Obstetrics. The results of the validation process and of the pilot study are presented and discussed.

In my opinion, information related to the development and validation of the questionnaire is mixed with data obtained from the population in which the pilot study has been carried out, for example, in part of the discussion I think it would be appropriate to eliminate the parts referring to the results of the pilot study. We thank the reviewer for his/her comment. We acknowledge that the main aim of this study was the development and validation of the questionnaire and the pilot study was part of the development process. Taking in account the suggestion made by the reviewer, we have better highlighted the outcomes from the pilot study to the questionnaire development, both in the Results and in the Discussion sections. At the same time, showing data from the study population and comparing them with national data on adult population and with other studies conducted among undergraduates, is necessary to understand if the questionnaire is working as we expect. Moreover, as far as we know, this is the first time that students from the Obstetric degree have been investigated about their smoking habits, so it also interesting showing the results in this regard. For these reasons, we show also the data obtained from the population. However, following the reviewer’s comment, the Results section regarding data from the study population has been shortened and clearly separated from the results regarding the questionnaire development. The same has been done in the Discussion.

In general, the text seems to me to be too long and poorly structured. I would suggest that the authors structure the content of the sections of their article along the lines of the one presented in this paragraph: "The questionnaire was developed according to the recommendation presented by the Association for Medical Education in Europe (AMEE) in the "Developing questionnaires for educational research: AMEE Guide No. 87" [14]. The Guide presents a systematic, seven-step process for designing high-quality questionnaires, including conduct a literature review, conduct interviews and/or focus groups, synthesize the literature review and interviews/focus groups, develop items, conduct expert validation, conduct cognitive interviews, and conduct pilot testing." We thank the reviewer for his/her comment. Following the reviewer’s suggestion, the paper has been restructured according to the AMEE guidelines. Moreover, the results section and the part of the discussion regarding the population study have been shortened.

Reviewer 2 Report

A clean and straightforward piece of research that opens the door for a better knowledge of the smoking habits and attitudes of the student population with a consequent reflection on the development of strategies for smoking cessation.

Maybe the inclusion of graphs and tables illustrating the results (or the most relevant) would be a plus (results section).

Author Response

A clean and straightforward piece of research that opens the door for a better knowledge of the smoking habits and attitudes of the student population with a consequent reflection on the development of strategies for smoking cessation. We thank the reviewer for his/her appreciation of our work.

Maybe the inclusion of graphs and tables illustrating the results (or the most relevant) would be a plus (results section). Thank you for your suggestion. A table illustrating the most relevant results has been added (Table 2). We report in Supplementary tables S1-S7 all results regarding the pilot study.

Reviewer 3 Report

I think your paper is well prepared for publication under section 3.2.2 you will see a small typo of "iss" instead of "is". Please correct it if I do not misread it.

Strength: The article contains as much information as possible, especially in the materials and method sections. Grammatical errors are rare throughout the paper, and the organization was beautiful. The author took time to prepare the report painstakingly. The introduction was straight and concise and kept on drawing readers to read on to the end.   Possible Improvement: The statistic section could benefit from a brief description of their relevance to the current work. Although this is not so relevant and in no way makes the paper of low quality. The result section contains several data in sentences; reported data/results could benefit from using a table.

Author Response

I think your paper is well prepared for publication under section 3.2.2 you will see a small typo of "iss" instead of "is". Please correct it if I do not misread it. Thank you, this has been corrected.

Strength: The article contains as much information as possible, especially in the materials and method sections. Grammatical errors are rare throughout the paper, and the organization was beautiful. The author took time to prepare the report painstakingly. The introduction was straight and concise and kept on drawing readers to read on to the end.  We thank the reviewer for his/her appreciation of our work.

 Possible Improvement: The statistic section could benefit from a brief description of their relevance to the current work. Although this is not so relevant and in no way makes the paper of low quality. The result section contains several data in sentences; reported data/results could benefit from using a table. Thank you for your suggestion. A brief description of the statistics used in this work has been added (par. 2.8). A table illustrating the most relevant results has been added (Table 2), while the results section has been shortened to avoid repetitions. We report in Supplementary tables S1-S7 all results regarding the pilot study.

Round 2

Reviewer 1 Report

Congratulations on your work
Best regards